# Probabilistic Time Series Forecasting with Structured Shape and Temporal Diversity

**Vincent Le Guen** [1,2]
vincent.le-guen@edf.fr

**Nicolas Thome** [2]
nicolas.thome@cnam.fr

[1] EDF R&D, Chatou, France
[2] Conservatoire National des Arts et Métiers, CEDRIC, Paris, France

## Abstract

Probabilistic forecasting consists in predicting a distribution of possible future outcomes. In this paper, we address this problem for non-stationary time series, which is very challenging yet crucially important. We introduce the STRIPE model for representing structured diversity based on shape and time features, ensuring both probable predictions while being sharp and accurate. STRIPE is agnostic to the forecasting model, and we equip it with a diversification mechanism relying on determinantal point processes (DPP). We introduce two DPP kernels for modeling diverse trajectories in terms of shape and time, which are both differentiable and proved to be positive semi-definite. To have an explicit control on the diversity structure, we also design an iterative sampling mechanism to disentangle shape and time representations in the latent space. Experiments carried out on synthetic datasets show that STRIPE significantly outperforms baseline methods for representing diversity, while maintaining accuracy of the forecasting model. We also highlight the relevance of the iterative sampling scheme and the importance to use different criteria for measuring quality and diversity. Finally, experiments on real datasets illustrate that STRIPE is able to outperform state-of-the-art probabilistic forecasting approaches in the best sample prediction.

## 1 Introduction

Time series forecasting consists in analysing historical signal correlations to anticipate future outcomes. In this work, we focus on probabilistic forecasting in non-stationary contexts, i.e. we aim at producing plausible and diverse predictions where future trajectories can present sharp variations. This forecasting context is of crucial importance in many applicative fields, e.g. climate [62, 34, 15], optimal control or regulation [66, 41], traffic flow [39, 38], healthcare [8, 1], stock markets [14, 7], *etc.* Our motivation is illustrated in the example of the blue input in Figure 1(a): we aim at performing predictions covering the full distribution of future trajectories, whose samples are shown in green.

State-of-the-art methods for time series forecasting currently rely on deep neural networks, which exhibit strong abilities in modeling complex nonlinear dependencies between variables and time. Recently, increasing attempts have been made for improving architectures for accurate predictions [31, 53, 37, 42, 35] or for making predictions sharper, e.g. by explicitly modeling dynamics [9, 16, 50], or by designing specific loss functions addressing the drawbacks of blurred prediction with mean squared error (MSE) training [12, 47, 33, 58]. Although Figure 1(b) shows that such approaches produce sharp and realistic forecasts, their deterministic nature limits them to a single trajectory prediction without uncertainty quantification.

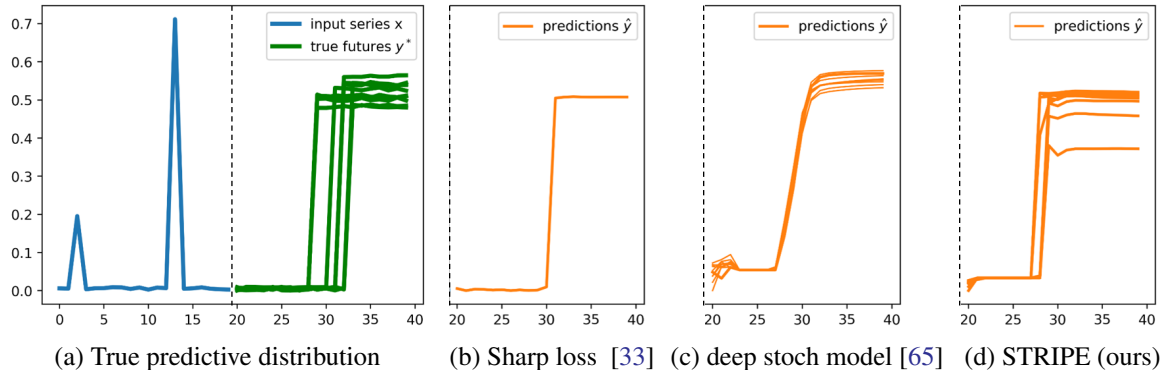

(a) True predictive distribution     (b) Sharp loss [33]   (c) deep stoch model [65]   (d) STRIPE (ours)

Figure 1: We address the probabilistic time series forecasting problem. (a) Recent deep learning models include a specific loss enabling sharp predictions [12, 47, 33, 58] (b), but are inadequate for producing diverse forecasts. On the other hand, probabilistic forecasting approaches based on generative models [65, 46] loose the ability to generate sharp forecasts (c). The proposed STRIPE model (d) produces both sharp and diverse future forecasts.

Methods targeting probabilistic forecasting enable to sample diverse predictions from a given input. This includes deterministic methods that predict the quantiles of the predictive distribution or probabilistic methods that sample future values from a learned approximate distribution, parameterized explicitly (e.g. Gaussian [52, 45, 51]), or implicitly with latent generative models [65, 29, 46]. These approaches are commonly trained using MSE or variants for probabilisting forecasts, e.g. quantile loss [28], and consequently often loose the ability to represent sharp predictions, as shown in Figure 1(c) for [65]. These generative models also lack an explicit structure to control the type of diversity in the latent space.

In this work, we introduce a model for including Shape and Time diverRsIty in Probabilistic forEcasting (STRIPE). As shown in Figure 1(d), this enables to produce sharp and diverse forecasts, which fit well the ground truth distribution of trajectories in Figure 1(a).

STRIPE presented in section 3 is agnostic to the predictive model, and we use both deterministic or generative models in our experiments. STRIPE encompasses the following contributions. Firstly, we introduce a structured shape and temporal diversity mechanism based on determinantal point processes (DPP). We introduce two DPP kernels for modeling diverse trajectories in terms of shape and time, which are both differentiable and proved to be positive semi-definite (section 3.1). To have an explicit control on the diversity structure, we also design an iterative sampling mechanism to disentangle shape and time representations in the latent space (section 3.2).

Experiments are conducted in section 4 on synthetic datasets to evaluate the ability of STRIPE to match the ground truth trajectory distribution. We show that STRIPE significantly outperforms baseline methods for representing diversity, while maintaining the accuracy of the forecasting model. Experiments on real datasets further show that STRIPE is able to outperform state-of-the-art probabilistic forecasting approaches when evaluating the best sample (i.e. diversity), while being equivalent based on its mean prediction (i.e. quality).

## 2 Related work

**Deterministic time series forecasting**    Traditional time series forecasting methods, including linear autoregressive models such as ARIMA [6] or exponential smoothing [27], handle linear dynamics and stationary time series (or made stationary by modeling trends and seasonality). Deep learning has become the state-of-the-art for automatically modeling complex long-term dependencies, with many works focusing on architecture design based on temporal convolution networks [5, 53], recurrent neural networks (RNNs) [31, 64, 44], or Transformer [57, 37]. Another crucial topic more recently studied in the non-stationary context is the choice of a suitable loss function. As an alternative to the mean squared error (MSE) largely used as a proxy, new differentiable loss functions were proposed to enforce more meaningful criteria such as shape and time [47, 12, 33, 58], e.g. soft-DTW based on

dynamic time warping [12, 4] or the DILATE loss with a soft-DTW term for shape and a smooth temporal distortion index (TDI) [20, 56] for accurate temporal localization. These works toward sharper predictions were however only studied in the context of deterministic predictions and not for multiple outcomes.

**Probabilistic forecasting** For describing the conditional distribution of future values given an input sequence, a first class of deterministic methods add variance estimation with Monte Carlo dropout [67, 32] or predict the quantiles of this distribution [61, 21, 60] by minimizing the pinball loss [28, 49] or the continuous ranked probability score (CRPS) [23]. Other probabilistic methods try to approximate the predictive distribution, *explicitly* with a parametric distribution (e.g. Gaussian for DeepAR [52] and variants [45, 51]), or *implicitly* with a generative model with latent variables (e.g. with conditional variational autoencoders (cVAEs) [65], conditional generative adversarial networks (cGANs) [29], normalizing flows [46]). However, these methods lack the ability to produce sharp forecasts by minimizing variants of the MSE (pinball loss, gaussian maximum likelihood), at the exception of cGANs - but which suffer from mode collapse that limits predictive diversity. Moreover, these generative models are generally represented by unstructured distributions in the latent space (e.g. Gaussian), which do not allow to have an explicit control on the targeted diversity.

**Diverse predictions** For improving the diversity of predictions, several repulsive schemes were studied such as the variety loss [26, 55] that consists in optimizing the best sample, or entropy regularization terms [13, 59] that encourage a uniform distribution and thus more diverse samples. Submodular distribution functions such as determinantal point processes (DPP) [30, 48, 40] are an appealing probabilistic tool to enforce structured diversity via the choice of a positive semi-definite kernel. DPPs has been successfully applied in various contexts, e.g. document summarization [24], recommendation systems [22], object detection [2], and very recently to image generation [17] and diverse trajectory forecasting [65]. GDPP [17] is based on matching generated and true sample diversity by aligning the corresponding DPP kernels, and thus limits their use in datasets where the full distribution of possible outcomes is accessible. In contrast, our approach is applicable in realistic scenarii where only a single label is available for each training sample. Although we share with [65] the goal to use DPP as diversification mechanism, the main limitation in [65] is to use the MSE loss for training the prediction and diversification models, leading to blurred prediction, as illustrated in Figure 1(c). Our approach is able to generate sharp and diverse predictions ; we also highlight the importance in STRIPE to use different criteria for training the prediction model (quality) and the diversification mechanism in order to make them cooperate.

## 3   Shape and time diversity for probabilistic time series forecasting

We introduce the STRIPE model for including shape and time diversity for probabilistic time series forecasting, which is depicted in Figure 2. Given an input sequence $\mathbf{x}_{1:T} = (\mathbf{x}_1, ..., \mathbf{x}_T) \in \mathbb{R}^{p \times T}$, our goal is to sample a set of $N$ diverse and plausible future trajectories $\hat{\mathbf{y}}^{(i)} = (\hat{\mathbf{y}}_{T+1}, ..., \hat{\mathbf{y}}_{T+\tau}) \in \mathbb{R}^{d \times \tau}$ from the data future distribution $\hat{\mathbf{y}}^{(i)} \sim p(.|\mathbf{x}_{1:T})$.

STRIPE builds upon a general Sequence To Sequence (Seq2Seq) architecture dedicated to multi-step time series forecasting: the input time series $\mathbf{x}_{1:T}$ is fed into an encoder that summarizes the input into a latent vector $h$. Note that our method is agnostic to the specific choice of the forecasting model: it can be a deterministic RNN, or a probabilistic conditional generative model (e.g. cVAE [65], cGAN [29], normalizing flow [46]).

For training the predictor (upper part in Figure 2), we concatenate $h$ with a vector $\mathbf{0}_k \in \mathbb{R}^k$ (free space left for the diversifying variables) and a decoder produces a forecasted trajectory $\hat{\mathbf{y}}^{(0)} = (\hat{\mathbf{y}}_{T+1}^{(0)}, ..., \hat{\mathbf{y}}_{T+\tau}^{(0)})$. The predictor minimizes a quality loss $\mathcal{L}_{quality}(\hat{\mathbf{y}}^{(0)}, \mathbf{y}^{(0)})$ between the predicted $\hat{\mathbf{y}}^{(0)}$ and ground truth future trajectory $\mathbf{y}^{(0)}$. In our non-stationary context, we train the STRIPE predictor with $\mathcal{L}_{quality}$ based on the recently proposed DILATE loss [33], that has proven successful for enforcing sharp predictions with accurate temporal localization.

For introducing structured diversity (lower part in Figure 2), we concatenate $h$ with diversifying latent variables $z \in \mathbb{R}^k$ and produce $N$ future trajectories $\{\hat{\mathbf{y}}^{(i)}\}_{i=1,...,N}$. Our key idea is to augment $\mathcal{L}_{quality}(\cdot)$ with a diversification loss $\mathcal{L}_{diversity}(\cdot; \mathcal{K})$ parameterized by diversity kernel $\mathcal{K}$ and

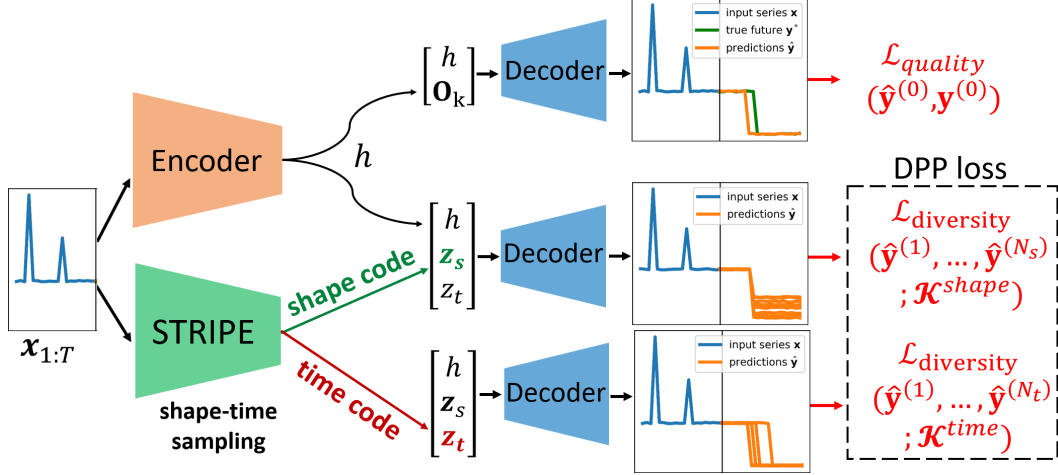

Figure 2: Our STRIPE model builds upon a Seq2Seq architecture trained with a quality loss $\mathcal{L}_{quality}$ enforcing sharp predictions. Our contributions rely on the design of a diversity loss $\mathcal{L}_{diversity}$ based on a specific Determinantal Point Processes (DPP). We design admissible shape and time DPP kernels, i.e. positive semidefinite, and differentiable for end-to-end training with deep models (section 3.1). We also introduce an iterative DDP sampling mechanism to generate disentangled latent codes between shape and time, supporting the use of different criteria for diversity and quality (section 3.2).

balanced by hyperparameter $\lambda \in \mathbb{R}$, leading to the overall objective training function:

$$\mathcal{L}_{STRIPE}(\hat{\mathbf{y}}^{(0)}, ..., \hat{\mathbf{y}}^{(N)}, \mathbf{y}^{(0)}; \mathcal{K}) = \mathcal{L}_{quality}(\hat{\mathbf{y}}^{(0)}, \mathbf{y}^{(0)}) + \lambda\, \mathcal{L}_{diversity}(\hat{\mathbf{y}}^{(1)}, ..., \hat{\mathbf{y}}^{(N)}; \mathcal{K}) \quad (1)$$

We highlight that STRIPE is applicable with a single target trajectory $\mathbf{y}^{(0)}$, i.e. we do not require the full trajectory distribution. We now detail how the $\mathcal{L}_{diversity}(\cdot; \mathcal{K})$ loss is designed to ensure diverse shape and time predictions.

## 3.1 STRIPE diversity module based on determinantal point processes

Our $\mathcal{L}_{diversity}$ loss relies on determinantal point processes (DPP) that are a convenient probabilistic tool for enforcing structured diversity via adequately chosen positive semi-definite kernels. For comparing two time series $\mathbf{y}_1$ and $\mathbf{y}_2$, we introduce the two following kernels $\mathcal{K}^{shape}$ and $\mathcal{K}^{time}$, for finely controlling the shape and temporal diversity:

$$\mathcal{K}^{shape}(\mathbf{y_1}, \mathbf{y_2}) = e^{-\gamma\, \text{DTW}_\gamma(\mathbf{y_1}, \mathbf{y_2})} \quad (2)$$

$$\mathcal{K}^{time}(\mathbf{y_1}, \mathbf{y_2}) = \text{TDI}_\gamma(\mathbf{y_1}, \mathbf{y_2}) = \frac{1}{Z} \sum_{\mathbf{A} \in \mathcal{A}_{\tau, \tau}} \langle \mathbf{A}, \mathbf{\Omega} \rangle \exp^{-\frac{\langle \mathbf{A}, \mathbf{\Delta}(\mathbf{y_1}, \mathbf{y_2}) \rangle}{\gamma}} \quad (3)$$

where $\text{DTW}_\gamma(\mathbf{y_1}, \mathbf{y_2}) := -\gamma \log \left( \sum_{\mathbf{A} \in \mathcal{A}_{\tau, \tau}} \exp^{-\frac{\langle \mathbf{A}, \mathbf{\Delta}(\mathbf{y_1}, \mathbf{y_2}) \rangle}{\gamma}} \right)$ is a smooth relaxation of Dynamic Time Warping (DTW) [12], and $\mathcal{K}^{time}$ corresponds to a smooth Temporal Distortion Index (TDI) [20, 33]: $\gamma > 0$ denotes the smoothing coefficient, $\mathbf{A} \subset \{0, 1\}^{\tau \times \tau}$ is a warping path between two time series of length $\tau$, $\mathcal{A}_{\tau, \tau}$ is the set of all feasible warping paths and $\mathbf{\Delta}(\mathbf{y_1}, \mathbf{y_2}) = [\delta((\mathbf{y_1})_i, (\mathbf{y_2})_j)]_{1 \le i, j \le \tau}$ is a pairwise cost matrix between time steps of both series with similarity measure $\delta : \mathbb{R}^d \times \mathbb{R}^d \to \mathbb{R}$, $\mathbf{\Omega}$ is a $\tau \times \tau$ matrix penalizing the deviation of warping paths from the main diagonal and $Z$ is the partition function. These kernels are derived from the two components of the DILATE loss [33] ; however in contrast to the deterministic nature of DILATE, they are used in a probabilistic context for producing sharp and diverse forecasts.

$\mathcal{K}^{shape}$ and $\mathcal{K}^{time}$ are differentiable by design[1], making them suitable for end-to-end training with back-propagation. We also derive the key following result for ensuring the submodularity properties of DPPs, that we prove in supplementary 1:

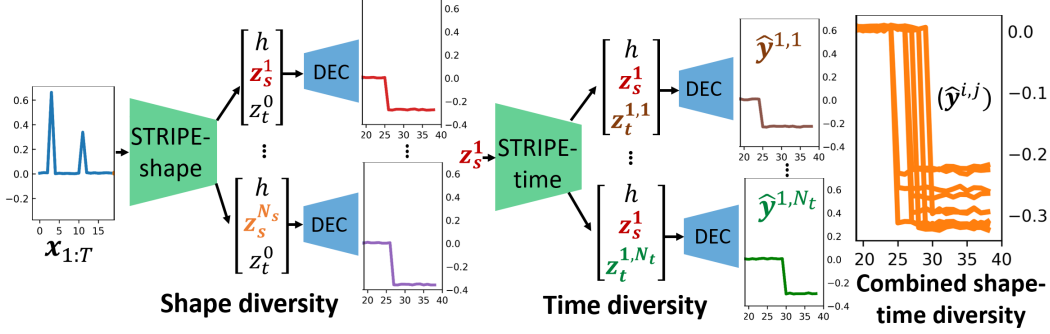

Figure 3: At test time, STRIPE sequential shape and time sampling scheme that leverages the disentangled latent space. STRIPE-shape first proposes diverse shape latent variables. For each generated shape, STRIPE-time further enhances its temporal variability, leading to a final set of accurate predictions with shape and time diversity.

**Proposition 1.** *Providing that $\kappa$ is a positive semi-definite (PSD) kernel $\kappa$ such that $\frac{\kappa}{1+\kappa}$ is also PSD, if we define the cost matrix $\Delta$ with general term $\delta(y_i, y_j) = -\gamma \log \kappa(y_i, y_j)$, then $\mathcal{K}^{shape}$ and $\mathcal{K}^{time}$ defined respectively in Equations (2) and (3) are PSD kernels.*

In practice, we choose $\kappa(u,v) = \frac{1}{2}e^{-\frac{(u-v)^2}{\sigma^2}}(1 - \frac{1}{2}e^{-\frac{(u-v)^2}{\sigma^2}})^{-1}$ that fullfills Prop 1 requirements.

**DPP diversity loss**   We combine the two differentiable PSD kernels $\mathcal{K}^{shape}$ and $\mathcal{K}^{time}$ with the DPP diversity loss from [65] defined as the negative expected cardinality of a random subset $Y$ (of a ground set $\mathcal{Y}$ of $N$ items) sampled from the DPP of kernel $\mathcal{K}$ (denoted as $\mathbf{K}$ in matrix form of shape $N \times N$). This loss is differentiable and can be efficiently computed in closed-form:

$$\mathcal{L}_{diversity}(\mathcal{Y}; \mathbf{K}) = -\mathbb{E}_{Y \sim DPP(\mathbf{K})}|Y| = -Trace(\mathbf{I} - (\mathbf{K} + \mathbf{I})^{-1}) \tag{4}$$

Intuitively, a larger expected cardinality means a more diverse sampled set according to kernel $\mathcal{K}$. We provide more details on DPPs and the derivation of $\mathcal{L}_{diversity}$ in supplementary 2.

### 3.2   STRIPE learning and sequential shape and temporal diversity sampling

To maximize shape and time diversity with Eq (1) and (4), a naive way is to consider the combined kernel $\mathcal{K}^{shape} + \mathcal{K}^{time}$ which is also PSD. However, this reduces to using the same criterion for quality and diversity, i.e. DILATE [33]. This directly makes $\mathcal{L}_{diversity}$ conflicts with $\mathcal{L}_{quality}$ and harms prediction performances, as shown in ablation studies (section 4.2). Another simple solution is to diversify using $\mathcal{K}^{shape}$ and $\mathcal{K}^{time}$ independently, which prevents from modeling joint shape and time variations, and intrinsically limits the expressiveness of the diversification scheme. In contrast, we propose a sequential shape and temporal diversity sampling scheme, which enables to jointly model variations in shape and time without altering prediction quality. We now detail how the STRIPE models are trained and then used at test time.

**STRIPE-shape and STRIPE-time learning**   We start by independently training two proposal modules STRIPE-shape and STRIPE-time (and their respective encoders and decoders) by optimizing Eq (1) with $\mathcal{L}_{STRIPE}(\cdot; \mathcal{K}^{shape})$ (resp. $\mathcal{L}_{STRIPE}(\cdot; \mathcal{K}^{time})$). To this end, we complement the latent state $h$ of the forecaster with a diversifying latent variable $z \in \mathbb{R}^k$ decomposed into shape $z_s \in \mathbb{R}^{k/2}$ and temporal $z_t \in \mathbb{R}^{k/2}$ components: $z = (z_s, z_t) \in \mathbb{R}^k$. As illustrated in Figure 3, STRIPE-shape (the description of STRIPE-time is symmetric) is a proposal neural network that produces $N_s$ different shape latent codes $z_s^{(i)}$ (the output of the STRIPE-shape neural network is of shape $N_s \times k$). The decoder takes the concatenated state $(h, z_s^{(i)}, z_t)$ for a fixed $z_t$ and produces $N_s$ future trajectories $\hat{y}^{(i)}$, whose diversity is maximized with $\mathcal{L}_{diversity}(\hat{\mathbf{y}}^{(1)}, ..., \hat{\mathbf{y}}^{(N_s)}; \mathbf{K}^{shape})$ in Eq (4). The architecture of STRIPE-time is similar to STRIPE-shape, except that the proposal neural network is conditioned on a generated shape variable $z_s^{(i)}$ to produce temporal variability with respect to a given shape.

**Sequential sampling at test time** Once the STRIPE-shape and STRIPE-time models (and their corresponding encoders and decoders) are learned, test-time sampling (illustrated in Figure 3 and detailed in Algorithm 1) consists in sequentially maximizing the shape diversity with STRIPE-shape (different guesses about the step amplitude in Figure 3) and the temporal diversity of each shape with STRIPE-time (the temporal localization of the step).

Notice that the ordering shape+time is actually important since the notion of time diversity between two time series is only meaningful if they have a similar shape (so that computing the DTW optimal path has a sense): the STRIPE-time proposals are conditioned on the shape proposals from the previous step. As shown in our experiments, this two-steps scheme (denoted STRIPE S+T) leads to more diverse predictions with both shape and time criteria compared to using the shape or time kernels alone.

---

**Algorithm 1:** STRIPE S+T sampling at test time

---

Sample an initial $z_t^{(0)} \sim \mathcal{N}(0, \mathbf{I})$
$z_s^{(1)}, ..., z_s^{(N_s)} =$
  STRIPE-shape$(\mathbf{x}_{1:T})$
**for** $i=1..N_s$ **do**
$\quad$ $z_t^{(i,1)}, ..., z_t^{(i,N_t)} =$
$\quad\quad$ STRIPE-time$(\mathbf{x}_{1:T}, z_s^{(i)})$
$\quad$ **for** $j=1..N_t$ **do**
$\quad\quad$ $\hat{\mathbf{y}}_{T+1:t+\tau}^{(i,j)} =$
$\quad\quad\quad Decoder(\mathbf{x}_{1:T}, (z_s^{(i)}, z_t^{(i,j)}))$
$\quad$ **end**
**end**

---

## 4 Experiments

To illustrate the relevance of STRIPE, we carry out experiments in two different settings: in the first one, we compare the ability of forecasting methods to capture the full predictive distribution of future trajectories on a synthetic dataset with multiple possible futures for each input. To validate our approach in realistic settings, we evaluate STRIPE on 2 standard real datasets (traffic & electricity) where we evaluate the best (resp. the mean) sample metrics as a proxy for diversity (resp. quality).

**Implementation details:** To handle the inherent ambiguity of the synthetic dataset (multiple targets for one input), our STRIPE model is based on a natively stochastic model (cVAE). Since this situation does not arise exactly for real-world datasets, we choose in this case a deterministic Seq2Seq predictor with 1 layer of 128 Gated Recurrent Units (GRU) [10]. In our experiments, all methods produce N=10 future trajectories that are compared to the unique (or multiple) ground truth(s). For a fair comparison, STRIPE S+T generates $N_s \times N_t = 10 \times 10 = 100$ predictions and we randomly sample N=10 predictions for evaluation. Further neural network architectures and implementation details are described in supplementary 3.1. Our PyTorch code implementing STRIPE is available at https://github.com/vincent-leguen/STRIPE.

### 4.1 Synthetic dataset with multiple futures

We use a synthetic dataset similar to [33] that consists in predicting step functions based on a two-peaks input signal (see Figure 1). For each input series of 20 timesteps, we generate 10 different future series of length 20 by adding noise on the step amplitude and localisation. The dataset is composed of $100 \times 10 = 1000$ time series for each train/valid/test split (further dataset description in supplementary 3.1).

**Metrics:** In this multiple futures context, we define two specific discrepancy measures $H_{quality}(\ell)$ and $H_{diversity}(\ell)$ for assessing the divergence between the predicted and true distributions of futures trajectories for a given loss $\ell$ ($\ell$ = MSE or DILATE in our experiments):

$$H_{quality}(\ell) = \mathbb{E}_{\mathbf{x} \in \mathcal{D}_{test}} \mathbb{E}_{\hat{\mathbf{y}}} \left[ \inf_{\mathbf{y} \in F(\mathbf{x})} \ell(\hat{\mathbf{y}}, \mathbf{y}) \right] \quad H_{diversity}(\ell) = \mathbb{E}_{\mathbf{x} \in \mathcal{D}_{test}} \mathbb{E}_{\mathbf{y} \in F(\mathbf{x})} \left[ \inf_{\hat{\mathbf{y}}} \ell(\hat{\mathbf{y}}, \mathbf{y}) \right]$$

$H_{quality}$ penalizes forecasts $\hat{\mathbf{y}}$ that are far away from a ground truth future of $\mathbf{x}$ denoted $\mathbf{y} \in F(\mathbf{x})$ (similarly to the precision concept in pattern recognition) whereas $H_{diversity}$ penalizes when a true future is not covered by a forecast (similarly to recall). We also use the continuous ranked probability score (CRPS)[2] which is a standard *proper scoring rule* [23] for assessing probabilistic forecasts [21].

Table 1: Forecasting results on the synthetic dataset with multiple futures for each input, averaged over 5 runs (mean $\pm$ standard deviation). Best equivalent method(s) (Student t-test) shown in bold. Metrics are scaled (MSE $\times$ 1000, DILATE $\times$100, CRPS $\times$ 1000) for readability.

| Methods | $H_{quality}$ $(.)(\downarrow)$ | | $H_{diversity}(.)$ $(\downarrow)$ | | CRPS $(\downarrow)$ |
|---|---|---|---|---|---|
| | MSE | DILATE | MSE | DILATE | |
| Deep AR [52] | $26.6 \pm 6.4$ | $67.0 \pm 12.0$ | $\mathbf{15.2 \pm 3.4}$ | $45.4 \pm 4.3$ | $62.4 \pm 9.9$ |
| cVAE MSE | $11.8 \pm 0.5$ | $48.8 \pm 3.2$ | $20.0 \pm 0.6$ | $85.4 \pm 7.0$ | $76.4 \pm 3.0$ |
| variety loss [55] MSE | $13.1 \pm 2.7$ | $50.9 \pm 4.7$ | $19.6 \pm 1.1$ | $84.7 \pm 2.2$ | $80.1 \pm 3.3$ |
| Entropy regul. [13] MSE | $12.0 \pm 0.7$ | $51.5 \pm 2.9$ | $19.7 \pm 0.7$ | $89.5 \pm 7.4$ | $78.9 \pm 2.9$ |
| Diverse DPP [65] MSE | $15.9 \pm 2.6$ | $56.6 \pm 2.8$ | $16.5 \pm 1.5$ | $59.6 \pm 5.6$ | $80.5 \pm 6.1$ |
| GDPP [17] kernel MSE | $11.7 \pm 1.3$ | $47.5 \pm 3.1$ | $19.5 \pm 0.4$ | $82.3 \pm 5.2$ | $74.0 \pm 4.5$ |
| **STRIPE S+T (ours)** | $12.4 \pm 1.0$ | $48.7 \pm 0.7$ | $18.1 \pm 1.6$ | $62.0 \pm 5.4$ | $72.2 \pm 3.1$ |
| cVAE DILATE | $11.6 \pm 1.8$ | $\mathbf{28.3 \pm 2.9}$ | $22.2 \pm 2.5$ | $67.8 \pm 7.8$ | $62.2 \pm 4.2$ |
| variety loss [55] DILATE | $14.9 \pm 3.3$ | $33.5 \pm 1.9$ | $23.8 \pm 3.9$ | $61.6 \pm 1.9$ | $62.6 \pm 3.0$ |
| Entropy regul. [13] DILATE | $12.7 \pm 2.6$ | $29.9 \pm 3.2$ | $23.5 \pm 2.6$ | $65.1 \pm 4.5$ | $62.4 \pm 3.9$ |
| Diverse DPP [65] DILATE | $\mathbf{11.1 \pm 1.6}$ | $30.2 \pm 2.9$ | $20.7 \pm 2.3$ | $62.6 \pm 11.3$ | $\mathbf{60.7 \pm 1.6}$ |
| GDPP [17] kernel DILATE | $\mathbf{10.6 \pm 1.6}$ | $\mathbf{28.7 \pm 4.1}$ | $21.7 \pm 2.1$ | $47.7 \pm 9.0$ | $63.4 \pm 6.4$ |
| **STRIPE S+T (ours)** | $\mathbf{10.8 \pm 0.4}$ | $30.7 \pm 0.9$ | $\mathbf{14.5 \pm 0.6}$ | $\mathbf{35.5 \pm 1.1}$ | $\mathbf{60.5 \pm 0.4}$ |

**Results** We compare our method to 4 recent competing diversification mechanisms (variety loss [55], entropy regularisation [13], diverse DPP [65] and GDPP [17]) based two different forecasting backbones: a conditional variational autoencoder (cVAE) trained with MSE and with DILATE. Results in Table 1 show that our model STRIPE S+T based on a cVAE DILATE obtains the global best performances by improving the diversity by a large margin ($H_{diversity}$(DILATE) = 35.5 vs. 67.8), significantly outperforming other methods. This highlights the relevance of the structured shape and time diversity in STRIPE. It is worth mentioning that STRIPE also presents the best performances in quality. In contrast, other diversification mechanisms (variety loss, entropy regularisation, diverse DPP) based on the same backbone (cVAE DILATE) improve the diversity in DILATE but at the cost of a loss in quality in MSE and/or DILATE. Although GDPP does not deteriorate quality, it is significantly worse than STRIPE in diversity, and the approach requires full future distribution supervision, which it not applicable in in real dataset (see section 2).

Similar conclusions can be drawn for the cVAE MSE backbone: the different diversity mechanisms improve the diversity but at the cost of a loss of quality. For example, Diverse DPP MSE [65] improves diversity ($H_{diversity}$(DILATE) = 59.6 vs. 85.4) but looses in quality ($H_{quality}$(DILATE) = 56.6 vs. 48.8). In contrast, STRIPE S+T again both improves diversity ($H_{diversity}$(DILATE) = 62.0 vs. 85.4) with equivalent quality ($H_{quality}$(DILATE) = 48.7 vs. 48.8). We further highlight that STRIPE S+T gets the best results evaluated in CPRS, confirming its ability to better recover the true future distribution.

## 4.2 Ablation study

To analyze the respective roles of the quality and diversity losses, we perform an ablation study on the synthetic dataset with the cVAE backbone trained with the quality loss DILATE and different DPP diversity losses. For a finer analysis, we report in Table 2 the shape (DTW, computed with Tslearn [54]) and time (TDI) components of the DILATE loss [33].

Table 2: Ablation study on the synthetic dataset. We train a backbone cVAE with the DILATE quality loss and compare different DPP kernels for diversity. Metrics are scaled for readability. Results averaged over 5 runs (mean $\pm$ std). Best equivalent method(s) (Student t-test) shown in bold.

| cVAE DILATE | $H_{quality}(.)$ $(\downarrow)$ | | $H_{diversity}(.)$ $(\downarrow)$ | | | | CRPS $(\downarrow)$ |
|---|---|---|---|---|---|---|---|
| diversity | MSE | DILATE | MSE | DTW | TDI | DILATE | |
| None | $11.6 \pm 1.8$ | $\mathbf{28.3 \pm 2.9}$ | $22.2 \pm 2.5$ | $18.8 \pm 1.3$ | $48.6 \pm 2.2$ | $67.8 \pm 7.8$ | $62.2 \pm 4.2$ |
| DILATE | $\mathbf{11.1 \pm 1.6}$ | $30.2 \pm 2.8$ | $20.7 \pm 2.3$ | $18.6 \pm 1.6$ | $42.8 \pm 10.2$ | $62.6 \pm 11.3$ | $60.7 \pm 1.7$ |
| MSE | $\mathbf{10.9 \pm 1.5}$ | $30.2 \pm 2.9$ | $20.1 \pm 2.2$ | $18.5 \pm 1.3$ | $41.9 \pm 8.8$ | $61.7 \pm 9.5$ | $62.1 \pm 0.9$ |
| shape (ours) | $\mathbf{11.0 \pm 1.4}$ | $30.2 \pm 1.2$ | $15.5 \pm 1.04$ | $\mathbf{16.4 \pm 1.5}$ | $15.4 \pm 4.2$ | $37.8 \pm 3.7$ | $63.2 \pm 1.6$ |
| time (ours) | $11.9 \pm 0.5$ | $31.2 \pm 1.3$ | $16.1 \pm 0.70$ | $17.6 \pm 0.5$ | $\mathbf{15.1 \pm 3.1}$ | $38.9 \pm 3.3$ | $62.3 \pm 1.4$ |
| **S+T (ours)** | $\mathbf{10.8 \pm 0.4}$ | $30.7 \pm 0.9$ | $\mathbf{14.5 \pm 0.6}$ | $\mathbf{16.1 \pm 1.1}$ | $13.2 \pm 1.7$ | $35.5 \pm 1.1$ | $60.5 \pm 0.4$ |

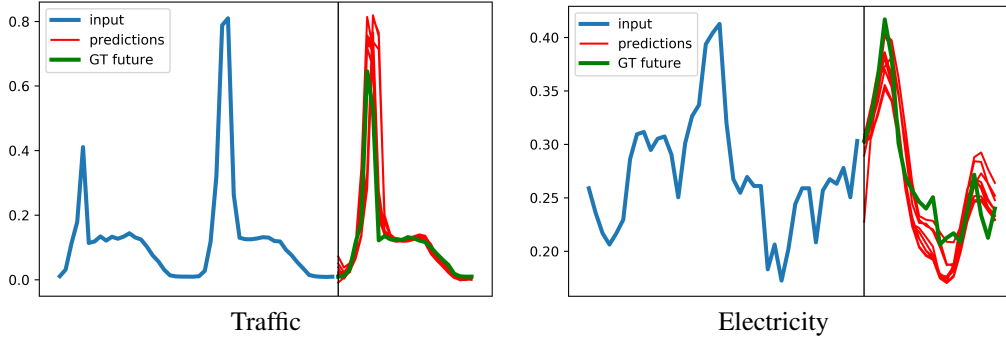

Figure 4: Qualitative predictions for Traffic and Electricity datasets. Input series in blue are not shown entirely for readability. We display 10 future predictions of STRIPE S+T that are both sharp and accurate compared to the ground truth (GT) future in green.

Results presented in Table 2 first reveal the crucial importance to define different criteria for quality and diversity. With the same loss for quality and diversity (as this is the case in [65]), we observe here that the DILATE DPP kernel does not bring a statistically significant diversity gain compared to the cVAE DILATE baseline (without diversity loss). By choosing the MSE kernel instead, we even get a small diversity and quality improvement.

In contrast, our introduced shape and time kernels $\mathcal{K}^{shape}$ and $\mathcal{K}^{time}$ largely improve the diversity in DILATE without deteriorating precision. As expected, each kernel brings its own benefits: $\mathcal{K}^{shape}$ brings the best improvement in the shape metric DTW ($H_{diversity}$(DTW) = 16.4 vs. 18.8) and $\mathcal{K}^{shape}$ the best improvement in the time metric TDI ($H_{diversity}$(TDI) = 15.1 vs. 48.6). With our sequential shape and time sampling sheme described in section 3.2, STRIPE S+T gathers the benefits of both criteria and gets the global best results in diversity and equivalent results in quality.

## 4.3 State-of-the-art comparison on real-world datasets

We evaluate here the performances of STRIPE on two challenging real-world datasets commonly used as benchmarks in the time series forecasting literature [63, 52, 31, 45, 33, 53]: **Traffic**: consisting in hourly road occupancy rates (between 0 and 1) from the California Department of Transportation, and **Electricity**: consisting in hourly electricity consumption measurements (kWh) from 370 customers. For both datasets, models predict the 24 future points given the past 168 points (past week). Although these datasets present daily, weakly, yearly periodic patterns, we are more interested here in modeling finer intraday temporal scales, where these signals present sharp fluctuations that are crucial for many applications, e.g. short-term renewable energy forecasts for load adjustment in smart-grids [34].

Contrary to the synthetic dataset, we only dispose of one future trajectory sample $\mathbf{y}^{(0)}_{T+1:T+\tau}$ for each input series $\mathbf{x}_{1:T}$. In this case, the metrics $H_{quality}$ (resp. $H_{diversity}$) defined in section 4.1 reduce to the mean sample (resp. best sample), which are common for evaluating stochastic forecasting models [3, 19]. We also report the CRPS in supplementary 3.2.

Results in Table 3 reveal that STRIPE S+T outperforms all other methods in terms of the best sample trajectory evaluated in MSE and DILATE for both datasets, while being equivalent in the mean sample in 3/4 cases. Interestingly, STRIPE S+T provides better best trajectories (evaluated in MSE

Table 3: Forecasting results on the Traffic and Electricity datasets, averaged over 5 runs (mean ± std). Metrics are scaled for readability. Best equivalent method(s) (Student t-test) shown in bold.

| | Traffic | | | | Electricity | | | |
| | MSE | | DILATE | | MSE | | DILATE | |
| Method | mean | best | mean | best | mean | best | mean | best |
|---|---|---|---|---|---|---|---|---|
| Nbeats [42] MSE | - | 7.8 ± 0.3 | - | 22.1 ± 0.8 | - | 24.6 ± 0.9 | - | 29.3 ± 1.3 |
| Nbeats [42] DILATE | - | 17.1 ± 0.8 | - | 17.8 ± 0.3 | - | 38.9 ± 1.9 | - | 20.7 ± 0.5 |
| Deep AR [52] | 15.1 ± 1.7 | **6.6 ± 0.7** | 30.3 ± 1.9 | 16.9 ± 0.6 | 67.6 ± 5.1 | 25.6 ± 0.4 | 59.8 ± 5.2 | 17.2 ± 0.3 |
| cVAE DILATE | **10.0 ± 1.7** | 8.8 ± 1.6 | **19.1 ± 1.2** | 17.0 ± 1.1 | **28.9 ± 0.8** | 27.8 ± 0.8 | 24.6 ± 1.4 | 22.4 ± 1.3 |
| Variety loss [55] | **9.8 ± 0.8** | 7.9 ± 0.8 | **18.9 ± 1.4** | 15.9 ± 1.2 | 29.4 ± 1.0 | 27.7 ± 1.0 | 24.7 ± 1.1 | 21.6 ± 1.0 |
| Entropy regul. [13] | 11.4 ± 1.3 | 10.3 ± 1.4 | **19.1 ± 1.4** | 16.8 ± 1.3 | 34.4 ± 4.1 | 32.9 ± 3.8 | 29.8 ± 3.6 | 25.6 ± 3.1 |
| Diverse DPP [65] | 11.2 ± 1.8 | 6.9 ± 1.0 | 20.5 ± 1.0 | 14.7 ± 1.0 | 31.5 ± 0.8 | 25.8 ± 1.3 | 26.6 ± 1.0 | 19.4 ± 1.0 |
| **STRIPE S+T** | **10.1 ± 0.4** | **6.5 ± 0.2** | **19.2 ± 0.8** | **14.2 ± 0.2** | 29.7 ± 0.3 | **23.4 ± 0.2** | 24.4 ± 0.3 | **16.9 ± 0.2** |

and DILATE) than the recent state-of-the-art N-Beats algorithm [42] (either trained with MSE or DILATE), which is dedicated to producing high quality deterministic forecasts. This confirms that STRIPE's structured quality and diversity framework enables to obtain very accurate best predictions. Finally when compared to the state-of-the art probabilistic deep AR method [52], STRIPE S+T is consistently better in diversity and quality.

We display a few qualitative forecasting examples of STRIPE S+T on Figure 4 and additional ones in supplementary 3.3. We observe that STRIPE predictions are both sharp and accurate: both the shape diversity (amplitude of the peaks) and temporal diversity match the ground truth future.

### 4.4 Model analysis

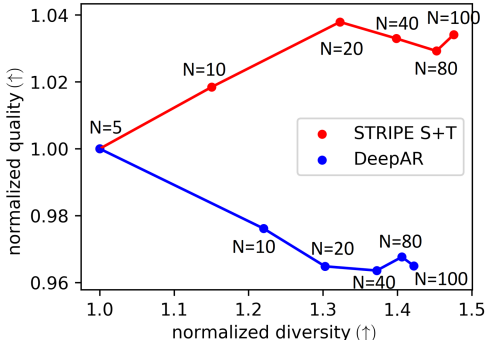

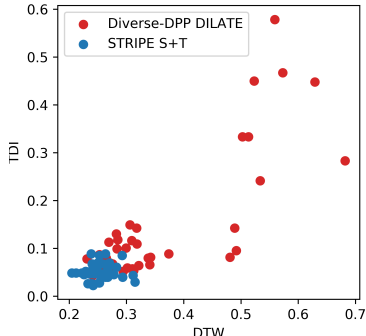

Figure 5: Influence of the number $N$ of trajectories on quality (higher is better) and diversity for the synthetic dataset.

Figure 6: Scatterplot of 50 predictions in the plane (DTW,TDI), comparing STRIPE S+T v.s. Diverse DPP DILATE [65].

We analyze in Figure 5 for the synthetic dataset the evolution of performances when increasing the number $N$ of sampled future trajectories from 5 to 100: we observe that this results in higher normalized DILATE diversity ($\mathrm{H}_{diversity}(5)/\mathrm{H}_{diversity}(N)$) for STRIPE S+T without deteriorating quality (which even increases slightly). In contrast, deepAR [52], which does not have control over the targetted diversity, increases diversity with $N$ but at the cost of a loss in quality. This again confirms the relevance of our approach that effectively combines an adequate quality loss function and a structured diversity mechanism.

We provide an additional analysis to highlight the importance to separate the criteria for enforcing quality and diversity. In Figure 6, we represent 50 predictions from the models Diverse DPP DILATE [65] and STRIPE S+T in the plane (DTW,TDI). Diverse DPP DILATE [65] uses a DPP diversity loss based on the DILATE kernel, which is the same than for quality. We clearly see that the two objectives conflict: this model increases the DILATE diversity (by increasing the variance in the shape (DTW) or the time TDI) components) but a lot of these predictions have a high DILATE loss (worse quality). In contrast, STRIPE S+T predictions are diverse in DTW and TDI, and maintain an overall low DILATE loss. STRIPE S+T succeeds in recovering a set of good tradeoffs between shape and time leading a low DILATE loss.

## 5 Conclusion and perspectives

We present STRIPE, a probabilistic time series forecasting method that introduces structured shape and temporal diversity based on determinantal point processes. Diversity is controlled via two proposed differentiable positive semi-definite kernels for shape and time and exploits a forecasting model with a disentangled latent space. Experiments on synthetic and real-world datasets confirm that STRIPE leads to more diverse forecasts without sacrificing on quality. Ablation studies also reveal the crucial importance to decouple the criteria used for quality and diversity.

A future perspective would be to incorporate seasonality and extrinsic prior knowledge (such as special events) [32, 42] to better model the non-stationary abrupt changes and their impact on diversity and model confidence [11]. Other appealing directions include diversity-promoting forecasting for exploration in reinforcement learning [43, 18, 36], and extension of structured diversity to spatio-temporal or video prediction tasks [62, 19, 25].

## Broader Impact

Probabilistic time series forecasting, especially in the non-stationary contexts, is a paramount research problem with immediate and large impacts in the society. A wide range of sensitive applications heavily rely on accurate forecasts of uncertain events with potentially sharp variations for making crucial decisions: in weather and climate science, better anticipating floods, hurricanes, earthquakes or other extreme events evolution could help taking emergency measures on time and save lives; in medicine, better predictions of an outbreak's evolution is a particularly actual topic. We believe that introducing meaningful criteria such as shape and time, which are more related to application-specific evaluation metrics, is an important step toward more reliable and interpretable forecasts for decision makers.

## Footnotes

[1]In the limit case $\gamma \to 0$, $\text{DTW}_\gamma$ (resp. $\text{TDI}_\gamma$) recovers the standard DTW (resp. TDI).

[2]An intuitive definition of the CRPS is the pinball loss integrated over all quantile levels. The CRPS is minimized when the predicted future distribution is identical to the true future distribution.

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
