[Supplementary Material]

# Probabilistic Time Series Forecasting with Structured Shape and Temporal Diversity
# Supplementary material

## 1 Proof of Proposition 1

We define the following kernels for comparing two trajectories $\mathbf{y} \in \mathbb{R}^{d \times \tau}$ and $\mathbf{z} \in \mathbb{R}^{d \times \tau}$:

$$\mathcal{K}^{shape}(\mathbf{y}, \mathbf{z}) = e^{-\gamma \, \mathrm{DTW}_\gamma(\mathbf{y}, \mathbf{z})} \tag{1}$$

$$\mathcal{K}^{time}(\mathbf{y}, \mathbf{z}) = \frac{1}{Z} \sum_{\mathbf{A} \in \mathcal{A}_{\tau,\tau}} \langle \mathbf{A}, \mathbf{\Omega} \rangle \exp^{-\frac{\langle \mathbf{A}, \mathbf{\Delta}(\mathbf{y}, \mathbf{z}) \rangle}{\gamma}} \tag{2}$$

where $\mathrm{DTW}_\gamma(\mathbf{y_1}, \mathbf{y_2}) := -\gamma \log \left( \sum_{\mathbf{A} \in \mathcal{A}_{\tau,\tau}} \exp^{-\frac{\langle \mathbf{A}, \mathbf{\Delta}(\mathbf{y_1}, \mathbf{y_2}) \rangle}{\gamma}} \right)$.

**Proposition 1.** *Providing that $\kappa$ is a positive semi-definite (PSD) kernel $\kappa$ such that $\frac{\kappa}{1+\kappa}$ is also PSD, if we define the cost matrix $\Delta$ with general term $\delta(y_i, z_j) = -\gamma \log \kappa(y_i, z_j)$, then $\mathcal{K}^{shape}$ and $\mathcal{K}^{time}$ defined respectively in Equations (1) and (2) are PSD kernels.*

*Proof.* The proof for $\mathcal{K}^{shape}$ is a direct consequence of Theorem 1 in [CVBM07]. Under the conditions that $\kappa$ and $\frac{\kappa}{1+\kappa}$ are PSD kernels, Theorem 1 in [CVBM07] states that for any alignment $\pi = (\pi_1, \pi_2)$ that respects the warping conditions, the following kernel $K$ is also PSD:

$$\begin{aligned}
K(\mathbf{y}, \mathbf{z}) &:= \sum_\pi \prod_{i=1}^{|\pi|} \kappa \left( y_{\pi_1(i)}, z_{\pi_2(i)} \right) \\
&= \sum_\pi \prod_{i=1}^{|\pi|} \exp^{-\frac{\delta\left( y_{\pi_1(i)}, z_{\pi_2(i)} \right)}{\gamma}} \\
&= \sum_\pi \exp^{-\sum_{i=1}^{|\pi|} \frac{\delta\left( y_{\pi_1(i)}, z_{\pi_2(i)} \right)}{\gamma}} \\
&= \sum_{\mathbf{A} \in \mathcal{A}_{\tau,\tau}} \exp^{-\frac{\langle A, \Delta(\mathbf{y}, \mathbf{z}) \rangle}{\gamma}} \\
&= \exp^{-\gamma \, \mathrm{DTW}_\gamma(\mathbf{y}, \mathbf{z})} \\
&= \mathcal{K}^{shape}(\mathbf{y}, \mathbf{z})
\end{aligned}$$

Let $a_1, ..., a_N \in \mathbb{R}$ and $\mathbf{y}_1, ..., \mathbf{y}_N \in \mathbb{R}^{d \times \tau}$. If $\Omega$ is non-zero on the diagonal (*e.g.* $\Omega(a, b) = \mu + \frac{(a-b)^2}{k^2}$ with $\mu > 0$), then there exists $\varepsilon > 0$ such that $\frac{\langle \mathbf{A}, \Omega \rangle}{Z} \geq \varepsilon$ $\forall \mathbf{A} \in \mathcal{A}_{\tau, \tau}$. Then:

$$\sum_i \sum_j a_i a_j \, \mathcal{K}^{time}(\mathbf{y}_i, \mathbf{y}_j) = \sum_i \sum_j a_i a_j \, \frac{1}{Z} \sum_{\mathbf{A} \in \mathcal{A}_{\tau,\tau}} \langle \mathbf{A}, \Omega \rangle \exp^{-\frac{\langle \mathbf{A}, \Delta(\mathbf{y}_i, \mathbf{y}_j) \rangle}{\gamma}}$$

$$\geq \sum_i \sum_j a_i a_j \sum_{\mathbf{A} \in \mathcal{A}_{\tau,\tau}} \varepsilon \exp^{-\frac{\langle \mathbf{A}, \Delta(\mathbf{y}_i, \mathbf{y}_j) \rangle}{\gamma}}$$

$$= \varepsilon \sum_i \sum_j a_i a_j \, \mathcal{K}^{shape}(\mathbf{y}_i, \mathbf{y}_j) \geq 0$$

The last inequality holds since we have already proven that $\mathcal{K}^{shape}$ is a PSD kernel. This proves that $\mathcal{K}^{time}$ is a PSD kernel. $\qquad\square$

The particular choice $\kappa(u, v) = \frac{1}{2} e^{-\frac{(u-v)^2}{\sigma^2}} (1 - \frac{1}{2} e^{-\frac{(u-v)^2}{\sigma^2}})^{-1}$ fullfills Prop 1 requirements: $\kappa$ is indeed PSD as the infinite limit of a sequence of PSD kernels $\sum_{i=1}^{\infty} k^i = \frac{k}{1-k} = \kappa$, where $k$ is a halved Gaussian PSD kernel: $k(u, v) = \frac{1}{2} e^{-\frac{(u-v)^2}{\sigma^2}}$.

For this choice of $\kappa$, the corresponding parwise cost matrix writes

$$\delta(y_i, z_j) = \gamma \left[ \frac{(y_i - z_j)^2}{\sigma^2} - \log \left( 2 - e^{\frac{-(y_i - z_j)^2}{\sigma^2}} \right) \right]$$

## 2 Derivation of $\mathcal{L}_{diversity}$

Determinantal Point Processes (DPPs) [KT$^+$12] are a probabilistic tool for describing the diversity of a ground set of items $\mathcal{S} = \{\mathbf{y}_1, ..., \mathbf{y}_N\}$. Diversity is controlled via the choice of a positive semi-definite (PSD) kernel $\mathcal{K}$ for comparing items. A DPP is a probability distribution over all subsets of $\mathcal{S}$ that assigns the following probability to a random subset $\mathbf{Y}$:

$$\mathcal{P}_{\mathbf{K}}(\mathbf{Y} = Y) = \frac{\det(\mathbf{K}_Y)}{\sum_{Y' \subseteq \mathcal{S}} \det(\mathbf{K}'_Y)} = \frac{\det(\mathbf{K}_Y)}{\det(\mathbf{K} + \mathbf{I})} \tag{3}$$

where $\mathbf{K}$ denotes the kernel in matrix form and $\mathbf{K}_A$ is its restriction to the elements indexed by $A$ : $\mathbf{K}_A = [\mathbf{K}_{i,j}]_{i,j \in A}$.

Intuitively, a DPP encourages the selection of diverse elements from the ground set $\mathcal{Y}$. If $\mathcal{Y}$ is more diverse, a random subset $Y \sim DPP(\mathcal{K})$ sampled from the DPP will select more items, *i.e.* will have a larger cardinality. This idea is embedded into the diversity loss $\mathcal{L}_{diversity}$ proposed in [YK20]:

$$\mathcal{L}_{diversity}(\mathcal{K}) = -\mathbb{E}_{Y \sim DPP(\mathcal{K})}|Y| = -Trace(\mathbf{I} - (\mathbf{K} + \mathbf{I})^{-1}) \tag{4}$$

## 3 Experiments

### 3.1 Datasets and implementation details

**Synthetic dataset** We use a synthetic dataset similar to [LGT19] that consists in predicting sudden changes (step functions) based on a two-peaks input signal. For each time series, the 20 first timesteps are the inputs, and the last 20 steps the targets to forecast. In each series, the input range is composed of 2 peaks at random temporal positions $i_1$ and $i_2$ and random amplitudes $j_1$ and $j_2$ between 0 and 1, and the target range is composed of a step of amplitude $j_2 - j_1$ at stochastic position $i_2 + (i_2 - i_1) + randint(-3; 3)$. All time series are corrupted by an additive Gaussian white noise of variance 0.01.

The difference with [LGT19] is that for each input series, we generate 10 different future series of length 20 by adding noise on the step amplitude and localisation. The dataset is composed of $100 \times 10 = 1000$ time series for each train/valid/test split.

**Neural network architectures** For the synthetic dataset, we use a stochastic predictive model based on a conditional variational autoencoder (cVAE).The encoder of the cVAE is a RNN with 1 layer of 128 GRU units, followed by a MLP which outputs the mean and variance of the latent state Gaussian distribution. We fixed by cross-validation the size of the latent state to $k = 16$. The decoder is another RNN with $128 + 16 = 144$ GRU units responsible for producing the future trajectory.

For the real-world datasets, we use a deterministic predictive Seq2Seq model with 1 layer of 128 GRU units for the encoder, and $128 + 16 = 144$ units for the decoder.

In all experiments, the STRIPE-shape proposal module is composed of a RNN with a layer of 128 GRU units followed by an MLP with 3 layers of 512 neurons (with BatchNormalization and LeakyReLU activations) and a final linear layer to produce $N = 10$ latent codes of dimension $k/2 = 8$ (corresponding to the proposals for $z_s$ or $z_t$).

The STRIPE-time proposal module has a similar architecture except that as input to the MLP, we concatenate the $z_s$ variable (of dimension 8) to condition the time variables on the current shape variable.

**STRIPE hyperparameters** We cross-validated the relevant hyperparameters of STRIPE:

- $\lambda$ : tradeoff between $\mathcal{L}_{quality}$ and $\mathcal{L}_{diversity}$. When increasing $\lambda$ (see Figure 1), the diversity increases and stabilizes starting from $10^{-3}$, without loosing on quality. We fixed $\lambda = 1$ in all experiments.

- $k$: dimension of the diversifying latent variables $z$. This dimension should be chosen relatively to the hidden size of the RNN encoders and decoders (128 in our experiments). We fixed $k = 16$ in all cases.

- $N$: the number of future trajectories to sample. We fixed $N = 10$. We performed a sensibility analysis to this parameter in paper section 4.4.

For computing the DILATE loss, we used the parameters recommended in paper [LGT19] ($\gamma = 0.01, \alpha = 0.5$).

Figure 1: Influence of the hyperparameter $\lambda$ balancing $\mathcal{L}_{quality}$ and $\mathcal{L}_{diversity}$ for the synthetic dataset. Quality (resp. diversity) are represented by $-\mathrm{H}_{quality}(\text{DILATE})$ (resp. $-\mathrm{H}_{diversity}(\text{DILATE})$), higher is better. When $\lambda$ increases, diversity increases without deteriorating quality.

## 3.2 Full state-of-the-art comparison results

We provide here (Table 1) the full results of the state-of-the-art comparison (Table 3 in paper). We report the additional CRPS metric. We observe that STRIPE S+T obtains the best results evaluated in CRPS on the Electricity dataset (equivalent to DeepAR [SFGJ20]), and the second best results on the Traffic dataset (only behind DeepAR that is otherwise far worse in diversity and quality).

Table 1: Forecasting results on the Traffic and Electricity datasets, averaged over 5 runs (mean ± std). Metrics are scaled for readability. Best equivalent method(s) (Student t-test) shown in bold.

| | Traffic | | | | | Electricity | | | | |
| | MSE (× 1000) | | DILATE (× 100) | | CRPS | MSE | | DILATE | | CRPS |
| Method | mean | best | mean | best | | mean | best | mean | best | |
|---|---|---|---|---|---|---|---|---|---|---|
| Nbeats MSE [OCCB20] | - | 7.8 ± 0.3 | - | 22.1 ± 0.8 | 37.1 ± 0.9 | - | 24.6 ± 0.9 | - | 29.3 ± 1.3 | 36.3 ± 0.6 |
| Nbeats DILATE | - | 17.1 ± 0.8 | - | 17.8 ± 0.3 | 51.0 ± 2.6 | - | 38.9 ± 1.9 | - | 20.7 ± 0.5 | 47.5 ± 0.5 |
| Deep AR [? ] | 15.1 ± 1.7 | **6.6 ± 0.7** | 30.3 ± 1.9 | 16.9 ± 0.6 | **24.6 ± 1.1** | 67.6 ± 5.1 | 25.6 ± 0.4 | 59.8 ± 5.2 | 17.2 ± 0.3 | **34.5 ± 0.3** |
| cVAE DILATE | **10.0 ± 1.7** | 8.8 ± 1.6 | **19.1 ± 1.2** | 17.0 ± 1.1 | 34.4 ± 2.5 | **28.9 ± 0.8** | 27.8 ± 0.8 | 24.6 ± 1.4 | 22.4 ± 1.3 | 39.2 ± 0.5 |
| Variety loss [TB19] | **9.8 ± 0.8** | 7.9 ± 0.8 | **18.9 ± 1.4** | 15.9 ± 1.2 | 32.4 ± 1.4 | 29.4 ± 1.0 | 27.7 ± 1.0 | 24.7 ± 1.1 | 21.6 ± 1.0 | 39.5 ± 0.8 |
| Entropy regul. [DRBT19] | 11.4 ± 1.3 | 10.3 ± 1.4 | **19.1 ± 1.4** | 16.8 ± 1.3 | 37.0 ± 2.7 | 34.4 ± 4.1 | 32.9 ± 3.8 | 29.8 ± 3.6 | 25.6 ± 3.1 | 42.4 ± 2.3 |
| Diverse DPP [YK20] | 11.2 ± 1.8 | 6.9 ± 1.0 | 20.5 ± 1.0 | 14.7 ± 1.0 | 30.9 ± 2.0 | 31.5 ± 0.8 | 25.8 ± 1.3 | 26.6 ± 1.0 | 19.4 ± 1.0 | 36.6 ± 0.9 |
| **STRIPE S+T** | **10.1 ± 0.4** | **6.5 ± 0.2** | **19.2 ± 0.8** | **14.2 ± 0.2** | 29.8 ± 0.3 | 29.7 ± 0.3 | **23.4 ± 0.2** | **24.4 ± 0.3** | **16.9 ± 0.2** | **34.8 ± 0.4** |

## 3.3 Additional visus

Wee provide additional visualizations for the Traffic and Electricity datasets that confirm that STRIPE S+T predictions are both diverse and sharp.

### 3.3.1 Electricity

### 3.3.2 Traffic