[Reviews · NeurIPS 2020]

Review 1

Summary and Contributions: The paper proposes a new probabalistic method for predicting time-series data by combining a time-series specific loss function (DILATE) with a time-series specific diversification mechanism based on determinantal point processes. The diversification mechanism has two components corresponding to time and shape (similar as the DILATE function) that are treated independently in the sampling and in the optimization procedure. The paper is a bit more empirical than many NeurIPS papers, but comes along with a strong empirical evaluation with a few minors pointed out below. Unfortunately, the paper lacks in clarity and important informations with respect to reproducibility are not given, which would make it hard to impossible for me to re-implement the algorithm. This is a huge disadvantage of the paper as it contradicts its main benefits, and the reason why I tend to a "weak reject".

Strengths: The idea of combining a time-specific loss function (DILATE) with a time-specific diversification mechanisms has not been conducted before. While the math for combining the two is not in particular complicated, the authors spent great effort in devicing an apropriate learning scheme that is rewarded by good experimental results.

Weaknesses: The main weaknesses of the paper are the missing clarity and reproducibility as further outlined below. I found the baseline "MSE-loss + S+T Diversity Sampling" missing in Table 1.

Correctness: The paper appears to be sound. Minor: Is Eq. (4) correct? I think it should be - Tr(K * (K+I)^(-1)).

Clarity: The training scheme is not clear to me. Since the training scheme is one of the core contributions of the paper, it deserves more space in the paper and its implementation details should be given either in the Supplement or in a GitHub repository. I would also highly recommend adding pseudo-code of the training scheme, similar as to the sampling scheme. Majors: - Do you first train Eq. (1) with K_shaped as L_diversity-Loss and afterwards E1. (1) with K_time as diversity as depicted in Figure 1? Or do you learn The STRIPE-shape component directly by optimizing Eq. (4) as written in Section 3.2? The encoder is not mentioned in Section 3.2. - For optimizing the STRIPE-time component, do you use the time componenty only as shown in Figure 1 or the sequential sampling scheme in Section 3.2? - Is the ordering relevant? Why do you not alternate between the different objectives? Do you use the same optimization settings for the different objectives?

Relation to Prior Work: The authors give a good overview over related work. Minor: The diversity kernels are derived from the DILUTE-loss which is not imminent from first reading.

Reproducibility: No

Additional Feedback: ___________________________________________________________________ After rebuttal: Thanks for your detailed answer. I am satisfied with it and increased my score accordingly.


Review 2

Summary and Contributions: In this paper, the authors deal with the time-series forecasting problem, particularly focusing on the probabilistic setting where multiple future outcomes are estimated. In the introduction they clearly present the main drawbacks of methods available in the literature: deep learning-based models are accurate and can capture sharp variations w.r.t. the groundtruth, but they are not able to propose multiple and diverse outcomes for a given input time-series; probabilistic methods can effectively solve the diversity issue but lose the sharpness of the predicted outcomes, and do not have control over the diversity. The authors introduce a method, called STRIPE, to overcome these problems: they use a loss function based on determinantal point processes (DPP) which exploits two kernels (K_shape and K_time) purposefully designed for controlling the shape and temporal diversity; moreover since K_shape and K_time can not be simply added and optimized jointly, the authors introduce an iterative process to model independently the variations in shape and time. Then, they consider the DILATE quality loss and perform an ablation study of various diversity losses, and finally they perform a comparison with state-of-the-art techniques on both synthetic and real world datasets.

Strengths: Interesting approach which deals directly with problems present in previous works in the literature. The authors perform an ablation study which shows why their claims are correct. They achieve sensible improvements over sota for both synthetic and real-world datasets.

Weaknesses: My main concern regards the two components they introduce, K_shape and K_time, which are almost the same as the components used in DILATE [27].

Correctness: They claim that combining K_shape and K_time and optimizing them together for diversity harms the performance, and that the iterative process to optimize the two components separately works better, and they show so in the Ablation study. Moreover they also show in Table 3 that using their approach they obtain sensible improvements w.r.t. the previous state-of-the-art over real-world datasets.

Clarity: The paper is overall well written, although there are some parts which are more difficult to follow. Also, there are small typos related to the acronyms (DPP appears as DDP on line 91 and in Fig. 2, and Table 3 shows STRIATE in place of STRIPE). Eq. 2 introduces the K_shape term, yet looking at the formula the authors are using it looks the same as the shape term introduced in the DILATE loss function (Eq (2) in [27]). Similar story for the K_time term, Eq (3) (which is Eq (4) in DILATE paper). In Section 3.2, lines 142-143 say that combining K_shape and K_time leads to “using the same criterion for quality and diversity, i.e. DILATE”. So, the two kernels are simply the two components already published in [27] ? After rebuttal: the authors should probably explicitly mention that the two DPPs are inspired by [27] although differing in several ways, as they explained in the rebuttal.

Relation to Prior Work: The authors clearly discuss the drawbacks of the previous works (w.r.t. the task at hand) and how they resolve them using the loss function and the iterative process (Algorithm 1). The only part which is not really clear to me is to how K_shape and K_time are novel w.r.t. what is introduced in DILATE [27]. After rebuttal: now it is definetely more clear how their work differs wrt [27]. I think the authors should use their rebuttal to update their paper accordingly.

Reproducibility: Yes

Additional Feedback: After rebuttal: the auhors explained the differences w.r.t. [27] and I think those differences should be better outlined in the paper as well.


Review 3

Summary and Contributions: The paper introduce what they call a structured shape and temporal diversity to improve the capability of probabilistic deep learning models to address sharp changes in the forecasting task.

Strengths: The paper is built on top of clear and correct mathematical formulas that are able to deal with spatial and temporal issues to aid the issue that are common when learning a model with the mae loss. The method is agnostic wrt to the predictive model. empirical results showed an improvement over other sota methods. A good finding is that DPP kernels are proven to be PSD,

Weaknesses: The works lack novelty in the way that it address the problem, is an incremental work built on top of “Shape and Time Distortion Loss for Training DeepTime Series Forecasting Models” proposed in neurips 2019. Wrt that work they took the DILATE loss that was proposed and they just separated in temporal and spatial part. The problem is that itself is not a contribution because in the original paper the DILATE loss itself was built from the separate spatial and temporal losses that were than lately conjoined to built the final loss. Experiments could be improved, adding more qualitative results.

Correctness: The method is clear and correct

Clarity: The paper is well written and manages to express clearly the scope of the work

Relation to Prior Work: The paper seems an incremental work from “Shape and Time Distortion Loss for Training DeepTime Series Forecasting Models” proposed in neurips 2019. While the experiments show improvement wrt to sota the paper, the difference wrt this latter should be better explained UPDATE. The rebuttal of the authors completely satisfied me.

Reproducibility: Yes

Additional Feedback: Since the work lack much novelty in the way the problem is addressed I would suggest to do a more in depth experiments and to better research cases where the work perform better than sota and to better address why it perform better. More qualitative results have to be shown


Review 4

Summary and Contributions: This paper describes a probabilistic timeseries forecasting approach that leverages shape and structural diversity. The key idea is to capture shape diversity using a smooth relaxation of dynamic time warping distance and temporal diversity using a smooth temporal distortion index. The key innovation is in disentangling shape and time representation into latent spaces. The Determinantal point process kernels proposed in this paper are differentiable and positive semi-definite - and thus easy to train as shown by the experimental evaluation in the paper

Strengths: Use of shape and temporal diversity in probabilistic timeseries forecasting DPP kernels that are differentiable and provably positive semi-definite Good experimental evaluation on both real and synthetic datasets.

Weaknesses: Key prior work has on seasonality based timeseries modeling (such as Holt-Winters, BATS) have been not been considered. Both traffic and electricity patterns show multiple seasonalities (daily, weekly). Lack of consideration of extrinsic factors such as working day/holiday, outside temperature and humidity (which affect heating/cooling systems and thus electricity).

Correctness: Appears to be correct. The selected kernels are PSD. The experimental results show that compared to some state of art techniques such as DeepAR the proposed solution offers both high quality and diversity in forecasts.

Clarity: The paper is generally well written and easy to understand.

Relation to Prior Work: The evaluation does not consider seasonality based timeseries models (statistical models).

Reproducibility: Yes

Additional Feedback:


Review 5

Summary and Contributions: The paper introduces a novel methodology to balance diversity and quality of forecasts via shape and time features. The authors claim that this method is independent of the forecasting method (although I didn't fully understand this point). The authors provide a thorough ablation and simulation study and provide initial experiments on real-world data. The authors claim that the methodology is in particularly well-suited for non-stationary time series.

Strengths: The methodology is sound and described in sufficient detail. It is novel and compared well to the existing approaches. The experiments, excluding those on real-world data sets, are well-designed and drive the authors' claims well. The problem that the authors tackle is highly relevant and has many practical applications.

Weaknesses: The paper claims that STRIPE works well for non-stationary time series, however the data sets, in particular the real-world data sets, do not support this claim. The traffic and electricity data sets are among the most regular real-world data sets there are. I'd like to see an analysis of the non-stationarity of these data sets. The paper claims that STRIPE is independent of the forecasting method, but I had troubles following this argument. Indeed, it relies on seq2seq architecture, so I'm unsure how methods like Deep State Models could be used. I'd ask the authors to explain this better. I'd ask the authors to go over the references. E.g., [43] is not from ICML but rather the International Journal of Forecasting. Also, which implementation of DeepAR is used? To the best of my knowledge, a number of available implementations (on Amazon SageMaker, PytorchTS or on GluonTS) are available. l. 60: ARIMA has a natural probabilistic version, so the statement is unnecessarily strong.

Correctness: The authors didn't submit source code with the paper. I encourage the authors to do this in the future. N=10 (l.163) future trajectories is insufficient for meaningful comparisons. Why not choose N=100 at least?

Clarity: yes

Relation to Prior Work: yes

Reproducibility: No

Additional Feedback: Thank you for the authors for the careful answer to the reviewers concerns. I will continue to maintain my vote.

[Author Response · NeurIPS 2020]

1  We thank the reviewers for their meaningful and valuable comments, which help to improve the quality of our work.

**R1:** • **Training details:** your first interpretation is correct: as stated in paper l.149, we independently train STRIPE-shape (resp. STRIPE-time) by optimizing Eq. (1) with $\mathcal{L}_{quality} + \lambda\mathcal{L}_{diversity-shape}$ (resp. $\mathcal{L}_{quality} + \lambda\mathcal{L}_{diversity-time}$) with the same optimization setting. Both models have their own encoder, decoder and diversification mechanism. A variant consists in jointly learning the STRIPE-shape and STRIPE-time modules with the sequential sampling scheme in Algo. 1, which we experimentally found to be inferior, probably due to the increased optimization difficulty of the overall sequential model. Therefore, the sequential shape and time scheme in section 3.2 is only used at inference time. **We will be glad to clarify these details in the training scheme and diversity sampling earlier in section 3, to add pseudo-code to the final version if accepted. We will also publish our code on Github with pretrained models for reproducibility.** In the sequential sampling scheme, the ordering shape+time is actually important since the notion of time diversity between two time series is only meaningful if they have a similar shape (so that computing the DTW optimal path for has a sense). We do not alternate the objectives since this would result in a combinatorial increase in the number of trajectories, and we have a fixed budget of N trajectories during inference.

• **MSE-loss with S+T diversity:** we show below the requested results. It follows the general trends observed in the submission: STRIPE S+T leads to a large diversity gain compared to the baselines without deteriorating quality. We will be glad to add this result in Table 1 of the final paper if accepted.

| | $H_{quality}(MSE)$ | $H_{quality}(DILATE)$ | $H_{diversity}(MSE)$ | $H_{diversity}(DILATE)$ | CRPS |
|---|---|---|---|---|---|
| STRIPE S+T | $12.4 \pm 1.0$ | $48.7 \pm 0.7$ | $18.1 \pm 1.6$ | $62.0 \pm 5.4$ | $72.2 \pm 3.1$ |

• **Eq.(4):** Thank you for your thorough reading, we made a typo ($\mathcal{K}$ instead of $\mathbf{I}$ before the minus sign), which will be corrected. It is equivalent to your proposal since: $\mathcal{L}_{diversity}(\mathcal{K}) = -Tr(\mathbf{I} - (\mathcal{K} + \mathbf{I})^{-1}) = -Tr(\mathcal{K}(\mathcal{K} + \mathbf{I})^{-1})$.

**Paper novelty and differences to ref [27] (R2 & R3)**: our submission addresses the problem of probabilistic time series forecasting by looking for a **set of sharp and diverse predictions**. It is in sharp contrast with the deterministic context of DILATE [27]. Our main differences with [27] are as follows:

1. We propose a diversification mechanism based on DPPs. Although our shape and time diversity criteria are inspired from [27], **we design DPP kernels that we prove to be PSD**, an important requirement for the DPP framework. Note that this property is non-trivial since DTW is not a proper distance (triangle inequality not satisfied).

2. **R3:** We introduce a sequential sampling scheme to disentangle the shape and time diversity features. Ablation studies validate the relevance of this diversification procedure, and **bring a crucial take-home message on the importance of using different losses for prediction quality and diversity**.

Note that our DPP diversification method also paves the way to other PSD kernels based on shape and time (e.g. edit distance, signal correlation) but also to different diversity criteria, e.g. the autoregressive kernel [Cuturi & Doucet'11].

**R3 on experiments:** We conduct extensive experiments on synthetic and real-world datasets and detailed ablation studies that validate our claims, as highlighted by Reviewers 1,2,4,5. Note that Fig 1 in submission represents a comparative visualization obtained by running source code of deterministic [27] or probabilistic [54] baselines for the synthetic dataset. We will add comparative visualizations for the other datasets in supplementary if accepted.

**Non-stationarity and seasonality (R4 and R5):** Thank you for your relevant comments, Traffic and Electricity datasets indeed show daily, weakly, yearly periodic patterns. In this work, we are more interested in finer intraday temporal scales, where these signals can present sharp fluctuations that are challenging for many applications (e.g. short-term renewable energy forecasts for load adjustment in a smart-grid). We have not leveraged seasonality-based algorithms to keep the method generic. We can however note that we obtain better results (in paper Table 3) than N-Beats [33], a recent deep method which incorporates seasonality and trend models. Introducing seasonality and extrinsic features (such as special events) is an appealing future perspective to help models focus on the relevant non-stationary parts of these signals, as well as quantifying their impact on diversity.

**R5:** • **Choice of number $N$ of trajectories:** our motivation is to summarize the predictive distribution by providing a small set ($N = 10$) of diverse and probable scenarii to a decision-maker, especially in the context where sharp fluctuations can occur. Larger values of $N$ would be indeed useful to estimate the full predictive distribution and computing quantiles (which is not our main purpose here). However, note that in paper Fig 5, we show results of our approach STRIPE S+T compared to the strong DeepAR baseline [43] when $N$ increases from 10 to 100. In this case, STRIPE S+T still has a better quality and diversity than DeepAR: at $N = 100$, $H_{quality} = 30.3$, $H_{diversity} = 42.2$ for STRIPE S+T *vs.* 67.7 and 58.9 for DeepAR (lower is better) ; results for $N \in [10, 100]$ will be added in supplementary.

• **DeepAR implementation:** we use the PytorchTS code (which wraps the GluonTS implementation in Pytorch).

**Formatting :** Thank you, we will correct the acronym typos (R2), correct the DeepAR reference (R5) and check all other references.

[Meta-Review · NeurIPS 2020]

Two of five knowledgeable referees recommend acceptance and the remaining three give it marginal acceptance rating, and I also recommend acceptance. Regardless, we strongly urge you to revise your paper to address the key reviewer's questions and contents of your rebuttal, in particular but not limited to explaining the relationship of your work to [27].